# NetTCR-2.0 enables accurate prediction of TCR-peptide binding by using paired TCRα and β sequence data

Alessandro Montemurro[1], Viktoria Schuster[1], Helle Rus Povlsen[1], Amalie Kai Bentzen[2], Vanessa Jurtz[1], William D. Chronister[3], Austin Crinklaw [3], Sine R. Hadrup[2], Ole Winther [4,5,6], Bjoern Peters[3,7], Leon Eyrich Jessen [1] & Morten Nielsen [1,8 ✉]

Prediction of T-cell receptor (TCR) interactions with MHC-peptide complexes remains highly challenging. This challenge is primarily due to three dominant factors: data accuracy, data scarceness, and problem complexity. Here, we showcase that "shallow" convolutional neural network (CNN) architectures are adequate to deal with the problem complexity imposed by the length variations of TCRs. We demonstrate that current public bulk CDR3β-pMHC binding data overall is of low quality and that the development of accurate prediction models is contingent on paired α/β TCR sequence data corresponding to at least 150 distinct pairs for each investigated pMHC. In comparison, models trained on CDR3α or CDR3β data alone demonstrated a variable and pMHC specific relative performance drop. Together these findings support that T-cell specificity is predictable given the availability of accurate and sufficient paired TCR sequence data. NetTCR-2.0 is publicly available at https://services.healthtech.dtu.dk/service.php?NetTCR-2.0.

[1] Department of Health Technology, Section for Bioinformatics, Technical University of Denmark, DTU, 2800 Kgs, Lyngby, Denmark. [2] Department of Health Technology, Section for Experimental and Translational Immunology, Technical University of Denmark, DTU, 2800 Kgs, Lyngby, Denmark. [3] Center for Infectious Disease and Vaccine Research, La Jolla Institute for Immunology, La Jolla, CA 92037, USA. [4] Department of Biology, Bioinformatics Centre, University of Copenhagen, 2200 Copenhagen, Denmark. [5] Section for Cognitive Systems, Department of Applied Mathematics and Computer Science, Technical University of Denmark, Kgs., Lyngby 2800, Denmark. [6] Centre for Genomic Medicine, Rigshospitalet, Copenhagen University Hospital, København Ø 2100, Denmark. [7] Department of Medicine, Division of Infectious Diseases and Global Public Health, University of California, San Diego, La Jolla, CA 92037, USA. [8] Instituto de Investigaciones Biotecnológicas, Universidad Nacional de San Martín, Buenos Aires, Argentina. ✉email: morni@dtu.dk

T cells survey the health status of cells by scrutinizing their surface for the presence of foreign peptides presented in complex with major histocompatibility complex (MHC) molecules. This recognition by the T cell is facilitated by the T-cell Receptor (TCR). This crucial interaction between TCRs and peptide-MHC (pMHC) molecules thus forms a molecular switch defining a bottleneck for immune activation. Understanding the rules governing this interaction hence represents a paramount step in both personalized immune treatment and development of targeted vaccines.

The TCR is a heterodimeric protein, consisting of an α- and β-chain. The subpart of the TCR interacting with the pMHC complex is defined by six loops, three for each α- and β-chain. These loops determine the specificity of the TCR and are denoted complementarity determining regions (CDRs) 1–2–3. The current consensus is that the CDR3 loops primarily interact with the peptide, while the CDR1 and CDR2 loops interact with the MHC[1–3]. The peptide specificity is thus predominantly defined by the CDR3 loops. The diversity of the CDR3s is defined by the genomic recombination of the variable, diversity, and joining (VDJ) TCR-genes. However, while the α-chain is the result of a V- and J recombination, the β-chain contains the V-, D- and J genes creating a broader diversity. The result of this is that most data-generating studies have focused on the β-chain alone.

The majority of the publicly available TCR-pMHC-specificity data resides in the Immune Epitope Database (IEDB)[4], VDJdb[5], and McPAS-TCR[6], all of which primarily contain CDR3β-data. Several recent works have demonstrated the important shortcoming of this limited view on the TCR and demonstrated how the information on the specificity of the TCR toward its cognate pMHC target is carried by CDR3 of both α- and β-chains[7,8]. To investigate the pMHC specificity on paired α-/β-chains, single-cell (SC) technology is required. SC is considerably more costly, and thus much less paired-specificity data are publicly available. This is a critical shortcoming of current databases and highlights the urgent need for further development of cost-efficient SC technologies capable of accurate high-throughput paired-data generation[9].

While cost-efficient and accurate state-of-the-art high-throughput technologies for experimentally and computationally assessing the binding of a peptide to an MHC are available[10–12], for reasons explained above, the TCR component of the triad remains highly cost-intensive and low throughput and sparsely explored. This represents a major challenge in moving the field forward.

A number of studies have been published related to the prediction of TCR-pMHC interactions[7,13–21]. They present a wide range of data and modeling techniques. Most are constructed based on data from the IEDB, VDJdb, and/or McPAS-TCR and, in addition to the epitope information, make use of either CDR3β sequences alone[13–15], a mixture of CDR3α and CDR3β sequences[16], or smaller data sets entailing all 6 CDR3 sequences and potentially additional cellular information[17,18]. Methodologically, the different studies range from simple CDR3β alignment-based methods[19,22], over CDR similarity-weighted distances such as TCRdist[7], k-mer feature spaces in combination with PCA and decision trees (SETE[13]), random forests[20,21] such as TCRex[23], CNN-based (ImRex[16]), and Gaussian process classification methods (TCRGP[17]), to more complex approaches integrating natural language processing (NLP) methods (ERGO[14]). The overall conclusion from these earlier works is that while the prediction of TCR specificity is feasible, the volume and accuracy of current data limit the performance of the developed models. Moreover, these earlier works only to a limited extent address the high degree of redundancy present in TCR-interaction data sets,

making it difficult to assess the generalizability of the developed models.

We have earlier proposed a simple 1D CNN-based model, NetTCR-1.0[15], integrating peptide and CDR3β sequence information into a model for the prediction of TCR peptide specificity. Using a similar modeling framework, we here present an in-depth analysis of publicly available TCR-pMHC interaction data, with an emphasis on investigating the impact of data limitations and quality on model performance. Furthermore, the performance of the developed model is compared with simpler sequence-based models as well as more complex deep learning approaches and the impact of training on paired versus single-chain TCR-sequence data is investigated.

## Results

We set out to develop and benchmark models for the prediction of TCR-pMHC binding with a particular focus on investigating the quality of different data types, and the effect of using paired CDR3α/β versus CDR3β information only.

We started with data obtained from the IEDB, consisting of 9204 unique CDR3β sequences, each labeled to bind a single pMHC complex, and 387,598 negative data points derived from 10X single-cell sequencing (for details see "Materials and methods"). This data set is referred to as the β-chain data. Another, but smaller set of positive data points, was derived from combining IEDB and VDJdb data providing both CDR3α- and CDR3β-chain, leading to a paired chain set of 2744 unique TCR-peptide data points. The available data were highly heterogeneous in terms of studied peptides and HLA alleles with a majority (62%) of the IEDB data being restricted to HLA-A*02:01. Likewise, the vast majority of the HLA-A*02:01 restricted peptides were of length 9. Given this, for the further part of this work, we limited the analysis to HLA-A*02:01 and 9-mer peptides. Supplementary Fig. 1 presents TCR counts in the positive data sets for the three most abundant peptides NLVPMVATV (NLV) from human herpesvirus 5 (cytomegalovirus), GILGFVFTL (GIL) from influenza A virus, and GLCTLVAML (GLC) from human herpesvirus 4 (Epstein–Barr virus) in the two data sets. These three represent 99% and 92% of the β-chain and paired-chain data, respectively.

**Model performance: CDR3β data.** In a first attempt to evaluate the possibility of predicting TCR-peptide interactions, prediction models were constructed from the TCRβ data set. A critical part of the model development and evaluation relates to the procedure implemented for data preparation in the context of data redundancy and partitioning. Models were therefore trained and evaluated using cross-validation on different CDR3β data sets, characterized by different degrees of interpartitional redundancies. The performance was further evaluated on an external data set. For details on the data set preparation and interpartitional redundancies, refer to "Materials and methods". Here, two models were investigated, a sequence-similarity and a 1D CNN-based (NetTCR) model. The sequence-similarity-based model (baseline) serves here as a benchmark to investigate the added benefit of modeling the data using the more complex CNN framework. Performance of deeper and different neural network architectures was investigated subsequently. Cross-validation performance results as a function of the partitioning thresholds are shown in Fig. 1a. Here, the baseline model demonstrated the expected strong association between internal data redundancy and model performance, with a substantial and highly significant ($p < 0.0001$, bootstrap test with 10,000 replications) drop in performance as the partitioning threshold is decreased (from an AUC value of 0.67 at 99% to 0.63 at 90%)—resulting in a lower

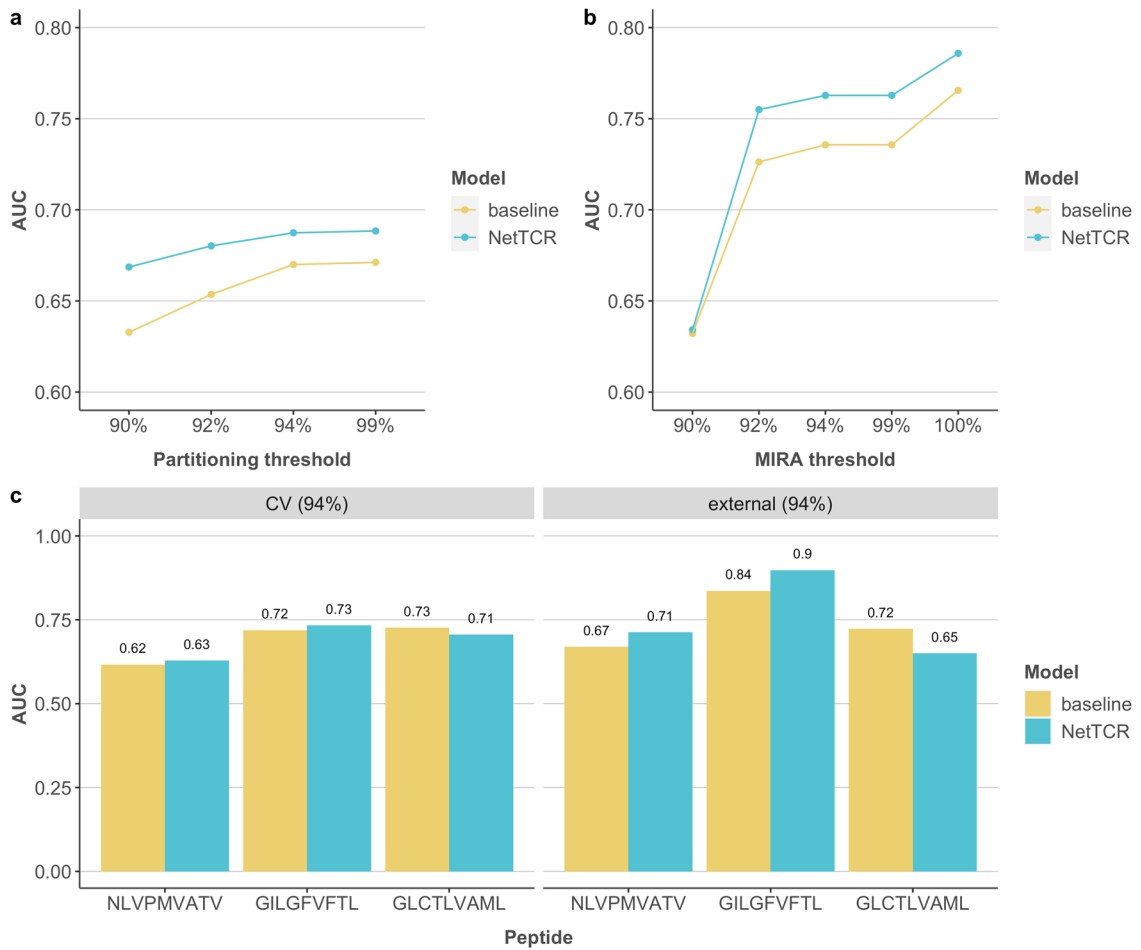

**Fig. 1 Performance of models trained on CDR3β data alone. a** Overall AUCs evaluated via cross-validation of different training data-partitioning thresholds for the baseline model and NetTCR. Partitioning thresholds are indicated in percent on the x-axis. **b** Overall AUCs evaluated on the MIRA sets at different thresholds (shown on the x-axis) using the model trained on the 94% similarity-partitioned data. The MIRA threshold represents the degree of separation between the training set and the MIRA set. **c** Peptide-specific AUCs for 94% partitioned cross-validation (CV) and external evaluation with a similarity threshold of 94%, colored by model.

similarity between the training and test data sets. This dependency on the partitioning threshold is diminished for the NetTCR neural network method. The performance of the NetTCR method was low even at the highest partitioning threshold with a maximum AUC of 0.69.

We next evaluated the performance of the models trained on the 94% partitioned data on the independent MIRA data set (Fig. 1b) using an ensemble of the 20 models obtained from cross-validation. Five different MIRA datasets were obtained by imposing a separation from the training set of 90, 92, 94, 99, and 100% similarity. That is, MIRA 94% TCRs do not share more than 94% Levenshtein similarity to any of the TCRs in the training set. Overall, this benchmark revealed a higher performance of all models compared to that observed in the cross-validation with a performance value of up to 0.79 in AUC. This performance is higher than the best-performance values observed during cross-validation and suggests that the MIRA data share an overall higher quality compared with the IEDB data used for training (for further discussion of this see later). Also here, the NetTCR method outperformed the baseline model, and we likewise observed a continued drop in performance of the models as the similarity between the evaluation and training data sets was diminished. This drop was particularly large for the 90% similarity threshold where all models achieved a comparable performance of AUC 0.635. Similar results were obtained for the

models trained using other partitioning thresholds (see Supplementary Fig. 2).

Figure 1c displays the peptide-specific AUCs in cross-validation and the external evaluation (defined using a 94% similarity threshold) of the models trained on the 94% partitioned training data set for the three dominant peptide sequences in the training data set. These peptide-specific AUCs strongly suggest that the model performance does not correlate with the amount of training data. That is, the performance of the NLV peptide characterized by the largest amount of training data displayed the lowest performance value in both the cross-validation and MIRA evaluation. Additionally, the neural network method did not in this evaluation perform overall better than the baseline model.

In conclusion, the observed relatively low predictive performance—even at high interpartitional redundancies—and the lacking correlation between data set size and predictive performance, suggest that TCR-peptide interactions can only to a very limited extent be characterized using current CDR3β-peptide data.

To further elaborate on this conclusion, and to ensure that it was not a result of the data set and/or modeling framework investigated here, we extended the benchmark to include the recently published ERGO method[14]. ERGO predicts peptide-TCR binding using long-short term memory (LSTM) networks or autoencoders (AE). Both network architectures were trained on

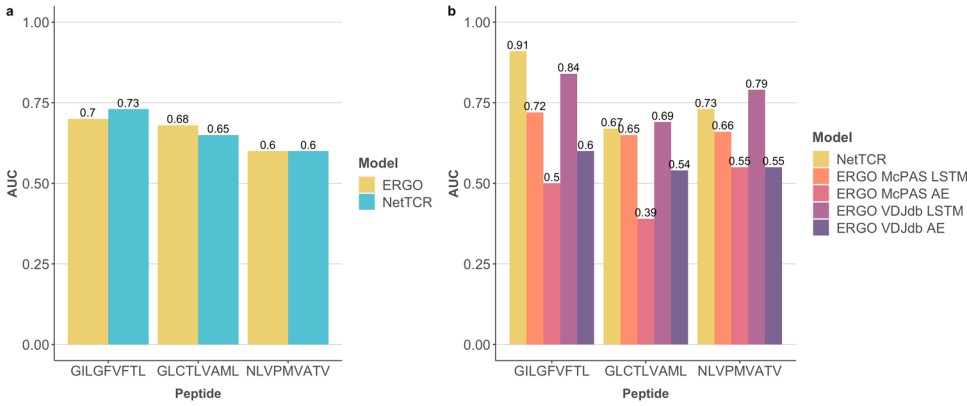

**Fig. 2 Comparison between NetTCR and ERGO. a** Test AUCs per peptide for NetTCR and ERGO trained on four out of five partitions of the IEDB + 10X data set and evaluated on the left-out partition. **b** Peptide-specific AUCs for NetTCR and all the four variants of ERGO evaluated on the MIRA data.

data sets derived from VDJdb and/or McPAS. Training NetTCR and the LSTM-based ERGO on four out of the five partitions of the IEDB + 10X data set and evaluating both models on the left-out partition, we observed that NetTCR and ERGO shared comparable performance in terms of peptide-specific AUC (see Fig. 2a) and both models have an overall AUC of 0.66. We further tested the performance of NetTCR trained on the complete IEDB + 10X data set and all the variants of ERGO on the MIRA data. In this case, NetTCR achieved an overall AUC of 0.77 and outperformed the best ERGO model (LSTM trained on VDJdb), which achieved an AUC of 0.74 (see Fig. 2b). These results show that NetTCR has a comparable performance to that of ERGO, hence demonstrating that the relatively low performance for TCR-peptide interactions observed here for NetTCR and the baseline is not imposed by the limited complexity of these models, compared with ERGO. Further, the results suggest that simple shallow models like the CNNs used here, rather than more sophisticated architectures, are sufficient to achieve optimal performance for the prediction of TCR-peptide specificity (at least given the current data).

**Model performance: paired CDR data**. Given the low performance of the CDR3β models, we next moved toward data sets consisting of both CDR3α and CDR3β. Figure 3a shows the overall and peptide-specific cross-validation AUC performance value of the baseline and NetTCR models trained on different TCR chain components for data sets created at 90% and 95% partitioning threshold. Here, data sets including both α- and β-chains, were partitioned by the average similarity of CDR3α and CDR3β. These partitions were maintained when training and evaluating models on α- or β-chains alone. The results from a chain-specific partitioning approach are included in Supplementary Fig. 3. These results in Fig. 3a demonstrate a comparable performance for models based on the CDR3α or CDR3β information and superior performance when including both the α- and β-CDR3 information for both the NetTCR and baseline models. With an overall AUC performance of 0.89, NetTCR significantly (p < 0.0001, bootstrap test with 10,000 replications) outperformed the baseline model. Further, the performance of the NetTCR model was found to be maintained when trained on the 90% compared with the 95% partitioned data. This was in contrast to the baseline model that suffered a significant drop in performance (p = 0.006, bootstrap test with 10,000 replications) when lowering the partition threshold. These observations are confirmed in Figs. 3b and 3c by the peptide-specific AUCs derived from the 90% and 95% partitioned data, respectively. Also here, and for both partitioning thresholds, the NetTCR model,

including both the α- and β-chain information, outperformed all other models, and both single-chain models achieved a lower but comparable performance. Investigating in more detail the effect of the size of the training data on the predictive performance of the two models, Fig. 3d displays the peptide-specific cross-validation AUC for the set of peptides included in the training data. Overall, this figure shows a decrease in AUC as the number of positive data points present in the training data drops, with an average AUC of NetTCR for peptides characterized by 200 or more TCRs of 0.88, and an average of peptides characterized by 20 or fewer TCRs of 0.38. One clear exception from this was the FLYALALLL peptide with only 37 binding TCRs and an AUC of 0.94. This potential outlier can however be explained by comparing the sequence similarities between positive and negative data points. Estimating a difference in similarity per positive TCR as the maximum similarity to all other positives for the given peptide in other partitions minus the maximum similarity to all negatives for the same peptide in other partitions, the expectation is that a higher dissimilarity between positives and negatives for a given peptide would ease the discrimination task, resulting in a higher peptide-specific performance value. This was confirmed by the result shown in Supplementary Fig. 4, where the AUC displays a clear tendency to increase as a function of the similarity difference (a Spearman correlation between AUC and median difference in similarity of 0.63). This result thus supports that FLYALALLL is an outlier and its high performance is imposed by the high difference in similarity score between its positive and negative TCRs.

Overall, these results suggest that consistent and high-performing models for TCR-pMHC interaction predicting can be developed from paired TCR data and that the low quality of current models is imposed by the low quality of bulk-sequenced CDR3β data. To further quantify this, we went back to the model trained on the bulk CDR3β data and evaluated using cross-validation the performance of a subset of 500 positive CDR3β shared with the paired TCR data sets, and an equal-size data set of positive CDR3β not sharing an overlap with the paired TCR data set. Both sets of positive TCRs were evaluated in the context of the complex negative dataset. The results of this experiment confirmed the high quality of the shared CDR3 data with an AUC of 0.80, and the likewise lower performance (AUC = 0.68) of the CDRs not shared with the single-cell data. Further, we evaluated the model trained on the 95% partitioned CDR3β data from the paired TCR data set on the CDR3β MIRA data (excluding identical overlap to the training data). This resulted in an overall AUC of 0.81. This performance is lower than the cross-validated performance but slightly higher than the performance of 0.79 demonstrated in Fig. 3b for the CDR3β-alone model. These results demonstrate that the MIRA data have a quality

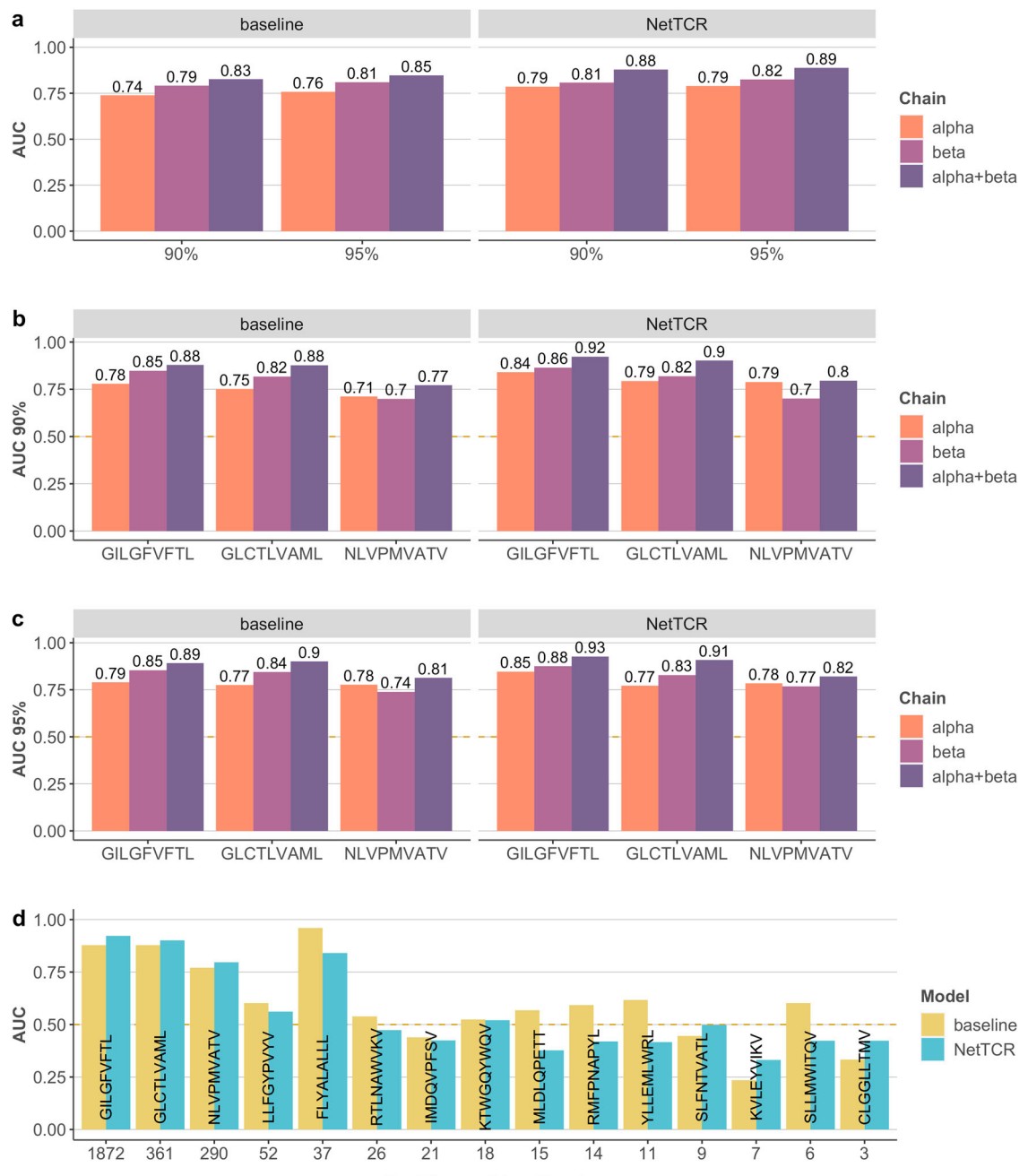

**Fig. 3 Performance of models trained on paired-chain data. a** Overall AUCs evaluated via cross-validation. **b**, **c** Peptide-specific AUCs from the 90% and 95% partitioned data for the three most frequent peptides. **d** Peptide-specific AUCs colored by model and plotted against the number of positive data points.

comparable to that of CDR3β from the paired TCR data, and thus, in line with the observation earlier, suggest a higher accuracy of these data compared with the overall accuracy of the bulk CDR3β alone data.

To further validate the high performance of the NetTCR-2.0 model, a performance comparison against TCRdist is included in Fig. 4 (for details on the implementation of the TCRdist method, refer to "Materials and methods"). This analysis aligns with the results from Fig. 3 demonstrating a consistent and highly significant ($p < 0.001$ for the α- and α + β-chain models, $p = 0.03$ for β-chain, bootstrap test with 1000 repetitions) superior performance of NetTCR-2.0 over TCRdist, and likewise showing that also for TCRdist is the signal in the CDR3β

sequence lower compared with CDR3α when it comes to predicting the specificity toward the NLV peptide.

Next, we investigated the power of the developed model to identify the correct peptide target of a given TCR. Here, binding to the three peptides GIL, NLV, and GLC was predicted (using cross-validation) for each TCR positive to any of these three peptides. To deal with peptide-specific scoring biases, the raw prediction values were transformed into the percentile rank values as described in "Materials and methods" and the predicted target for each TCR was identified from the peptide with the lowest rank value. This analysis was performed for the three models trained on the CDR3α and CDR3β, CDR3α alone and CDR3β alone, and the performance for each peptide was reported

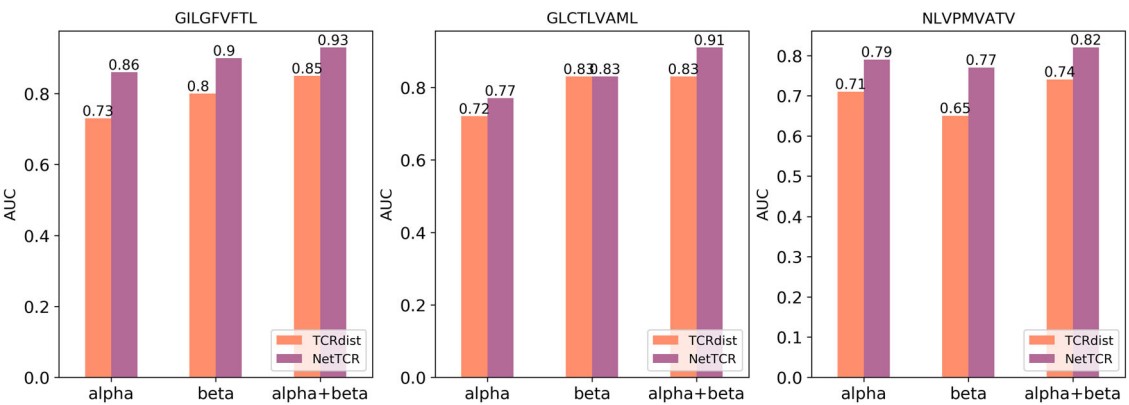

**Fig. 4 Comparison between NetTCR and TCRdist.** Performance is evaluated via cross-validation on the 95% partitioned data for the three most frequent peptides.

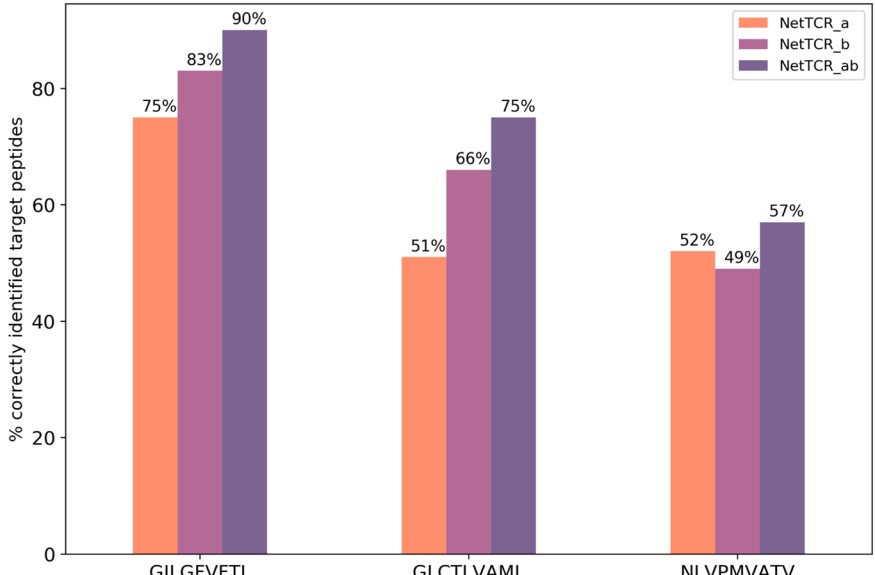

**Fig. 5 Peptide-ranking analysis.** Each TCR positive to GIL, GLC, or NLV peptide was paired to the other two peptides and a binding prediction was obtained. The percentages show, for each peptide and for each model, the proportion of TCRs for which the predicted lowest-ranking peptide matched with the "true" target peptide.

as the proportion of correctly identified targets (see Fig. 5). Here, all models performed better than random with the proportion of correct targets >33%. Further, the model trained on both CDR3α and CDR3β significantly outperformed both other models for all three peptides (p-value < 0.05 in all the cases, bootstrap test with 1000 repetitions); meanwhile, the choice of the best single-chain model was peptide dependent, with NetTCR_α outperforming NetTCR_β for the NLV peptide, in line with the result of Fig. 3. To further quantify to what extent the peptide sequence contributes to the model performance, models were trained on a data set where the TCR sequences were paired with a wrong peptide. Repeating the peptide-ranking analysis with these models demonstrated a highly reduced performance, exemplified with, for instance, the TCR_αβ for all TCRs predicting the optimal target as the GIL peptide (see Supplementary Fig. 5).

We propose that the improved predictive power of NetTCR over the sequence-based baseline model is driven by the representation of the TCRs in the max-pooled CNN layer of NetTCR. To elucidate this, the 160-dimensional representation max-pooled output (80 for each of the CDR3α and CDRb TCR sequences, respectively) from the NetTCR CNN layer of the CDR3α and CDR3β input was extracted for all TCRs specific to

the GIL peptide. Likewise, a raw input representation of the TCR was constructed using a simple encoding scheme where each amino acid was represented by five features (normalized Van der Waals volume, hydrophobicity, number of hydrogen bond donors, number of hydrogen bond acceptors, and net charge). Next, the t-distributed stochastic neighbor embedding (t-SNE[24]) algorithm was used to visualize the relationship between these vectors in a 2-dimensional space (see Fig. 6). In contrast to the raw sequence representation (Fig. 6b), Fig. 6a shows the separation of the positive from the negative GIL TCRs with a clear positive TCR-enriched region in the upper-left part.

To further illustrate how the max-pooled feature space allows for separation of the positive from the negative GIL TCRs, Fig. 7 shows a hierarchically clustered heatmap of a random set of 50 positive and 50 negative GIL TCRs. This figure clearly illustrates the increased power for separation of the positive from the negative TCR when information from both CDR3α and CDR3β is included. Further comparing the results obtained using the paired-chain max-pooled representation (Fig. 7a) to the raw input space (Supplementary Fig. 6), confirmed the improved clustering potential of the max-pooled sequence representation. To further quantify the increased ability of classification in the CNN space, the positive and negative

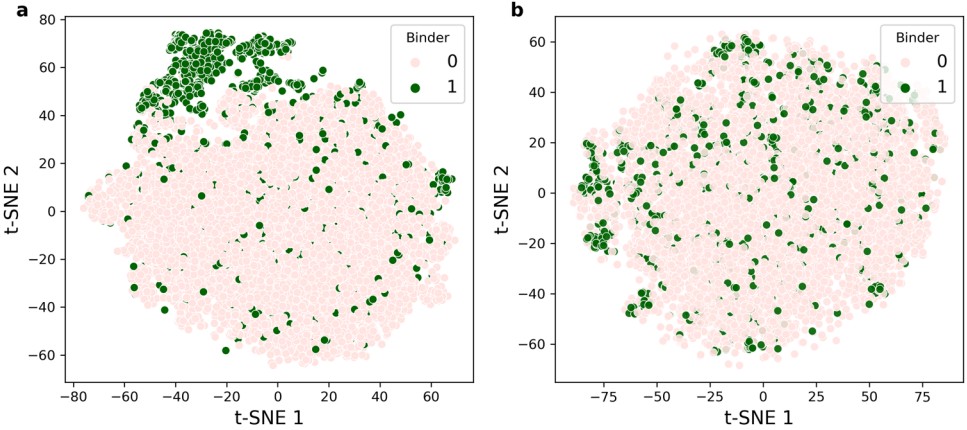

**Fig. 6 t-SNE plot for the TCRs of the GIL peptide. a** The output from the max-pooled CNN layer of NetTCR trained on the 90% partitioned data set was extracted for each TCR specific to the GIL peptide using cross-validation, resulting in a set of vectors, each of dimension 160. T-SNE was used to visualize this data set in two dimensions. **b** In the input space, the TCRs were encoded using a 5-feature physicochemical encoding and then flattened into a vector. The perplexity hyperparameter of the t-SNE algorithm was chosen to be 40 and the number of iterations was set to 1000. In the plot, positive TCRs are shown in green, and negative TCRs in pink.

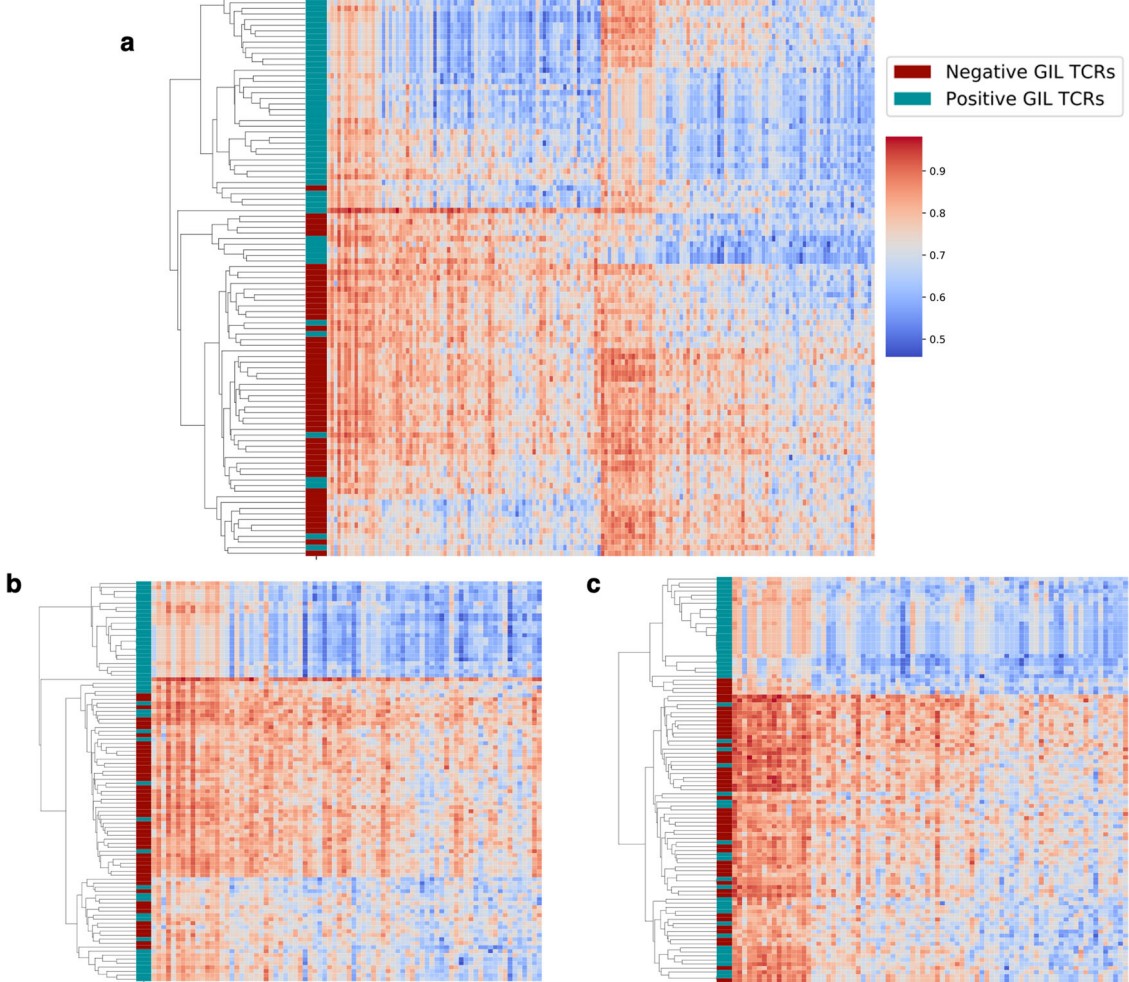

**Fig. 7 Hierarchical-clustered heatmaps of 50 positive GIL TCRs and 50 negatives.** The clustering was performed using both α- and β-sequences (**a**) or using single chains (α chain in **b**, β chain in **c**). Each row in the heatmap represents a TCR sequence in the max-pooled feature-space representation; the color bar on the side of each plot delineates whether the TCR is positive or negative. Cosine distance was used as a metric for clustering.

TCRs were clustered into two groups using the K-medoids algorithm. The two clusters were labeled as positive and negative by the majority vote of the TCRs falling in the cluster, and the clustering accuracy was evaluated using the Matthews correlation coefficient (MCC). The clustering was performed using both the max-pooled and the raw input representation of the TCRs, resulting in MCC values of $0.64 +/- 0.09$ and $0.21 +/- 0.14$ (standard-deviation values obtained using 1000 resamplings of TCR), confirming that the separation between positives and negatives is significantly more pronounced in the CNN space.

**The NetTCR server**. The presented NetTCR method is available as a web server at https://services.healthtech.dtu.dk/service.php? NetTCR-2.0. The server offers the possibility of predicting binding of the input TCRs with one or more peptides; predictions are made using the models trained on the 95% partitioned training data. Supplementary Fig. 7a, b serves as a guide to select thresholds for interpretation of prediction scores that the server outputs, and displays sensitivity–specificity curves of the method for the three individual peptides and the pooled data set with prediction values obtained as percentile rank scores using cross-validation. These figures demonstrate the very high specificity of the method with sensitivity values greater than 50% (and in most cases greater than 75%) and false-positive rates less than 2% in all cases using a percentile rank score threshold of 2%.

**Real-life validation**. As a real-life validation of the NetTCR-2.0 method, a performance comparison of the different models was conducted on a novel independent paired TCR data set generated specifically for this study. In short, the data were defined from T cells from four HLA-A*02:01-positive donors with pre-established responses to GILGFVFTL, NLVPMVATV, and GLCTLVAML sorted into a positive subset, containing TCRs responsive to one or more of the three peptides and a negative subset, containing TCRs negative to the three peptides. Here, the performance was estimated by predicting for each TCR binding to the three peptides and assigning a score corresponding to the lowest-predicted rank value. Next, performance values were calculated in terms of AUC, AUC0.1 (defined as the area under the ROC curve in the interval [0, 0.1]), and positive predictive value (PPV), calculated as the proportion of positive hits within the top 89 (the total number of positive TCR) predicted TCR. Here, the performance measures were used to quantify how this prediction score could be used to separate the positive and negative TCRs (see Fig. 8). Also in this benchmark, NetTCR_$\alpha\beta$ significantly outperform all other methods ($p < 0.05$, bootstrap test with 10000 repetitions), with a performance gain of more than 10% in terms of PPV. Here the method demonstrate a very high specificity, identifying 79% of the positive TCR at a false-positive rate of 2% using a percentile rank threshold of 2% (Supplementary Fig. 7c).

## Discussion

Identification of cognate targets of TCRs is a critical bottleneck of the development of T-cell therapeutics. Here, we have presented a study aiming to resolve this bottleneck, developing models capable of predicting TCR-pMHC interactions based on the amino acid sequences of the peptide and CDR3 region of the TCR chains. Several model architectures were investigated spanning from simple sequence-similarity models to more complex convolutional neural networks (CNN). The models were trained using cross-validation and validated using independent evaluation data carefully constructed using strict data-redundancy reduction rules. The overall best-performing model was found to be a 1D CNN. This model is a variant of the model proposed earlier by us for pan-specific prediction of kinase-specific

phosphorylation[25]. This model significantly outperformed simpler sequence-based models implemented using the TCRMatch[22] and TCRdist[7] frameworks.

Two important issues related to the understanding of the TCR-binding characterization and prediction were addressed during the model development, namely the quality of the current data, and the impact of including paired CDRα and CDRβ information. First, models were developed using data available from the IEDB (similar results were obtained using CDR3β data from VDJdb) with CDR3β information available only. This data set was substantially larger compared with data with paired TCR-sequence information, and one would expect that models trained on such larger data sets should achieve overall higher performance values compared with models trained on the more reduced paired TCRα and TCRβ data sets. This was however not the case. Models constructed from data with CDR3β information from paired TCR data demonstrated significantly higher performance to similar models trained on the data with CDR3β information only. This result strongly suggests that the quality of the data with only CDR3β information is lower than that of the data with paired CDRs. Further, and in line with earlier work[7,8], the conclusions from the current study clearly supported the notion that both TCR chains contribute to the TCR specificity (and importantly, that their relative importance is pMHC specific), and that only by including this combined information can one achieve accurate TCR-specificity prediction.

In contrast to the models trained on the data with only CDR3β information, the model trained on the data with paired TCR information demonstrated a clear and statistically significant correlation of the peptide-specific performance to the number of different positive TCR available for a given peptide and suggested that ~150 unique TCRs are required to achieve an AUC > 0.75 for a given peptide. Currently, this criterion is only met for a very small set of MHC-peptide combinations placing great limitations on the applicability of the developed model, since it can only, given the current data, provide reliable predictions for three peptides. This limitation underlines the urgent need for the development and refinement of technologies for high-throughput paired sequencing of TCRs with known pMHC targets. The developed framework is trivially extendable and retrainable, as more data become available.

Investigating the TCR-specific performance of the model revealed a likewise high predictive power, with ~75% of predicted peptide targets (from the pool of three) being correct. Taken with some reservations, given the small peptide space covered, this high performance suggests that the model has the potential to resolve not only which TCRs are specific to a given peptide, but also which peptide is specific for a given TCR, pointing to important biomedical applications within T-cell therapy[26,27].

The power of the CNN model compared with the simpler sequence-based approaches lies in its ability to translate the variable length of the TCR sequences into an abstract feature space suitable for specificity classification. To illustrate this, a similarity analysis between TCRs specific to the GIL peptide was conducted in the CNN feature space compared with the original sequence space. This analysis confirmed the improved ability to perform classification in the CNN feature space and suggests that this representation potentially could be used as an alternative to the conventional autoencoding approaches for feature extraction and compression of biological data[28,29].

The current model only includes information from the two CDR3 regions of the TCR. Earlier work has demonstrated that also CDR1 and CDR2 carry information of potential importance for prediction of TCR specificity[7,8]. The modeling framework proposed here can readily be extended to include such information (as well as information related to HLA and V- and

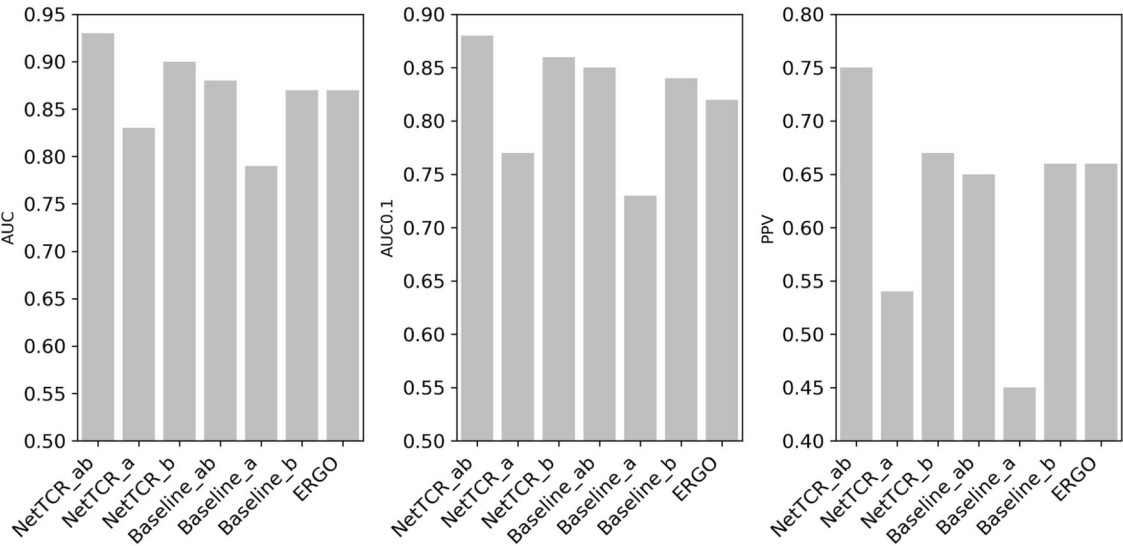

**Fig. 8 Benchmark performance on in-house TCR data set.** Methods included are NetTCR and baseline trained on paired CDR3α–CDR3β data (ab), CDR3α (a), CDR3β (b), and the LSTM-based ERGO trained on the VDJdb. Performance measures are (left) AUC, center (AUC 0.1), and right (PPV).

J- germline usage), and future work will tell if integrating this information can lead to an improved predictive power of the model proposed here. Also, the neural network architecture proposed here is relatively simple, consisting of one single max-pooled CNN layer. In this work, we did not perform an exhaustive performance comparison to other more complex models; however, our comparison to the ERGO model on the CDR3β-only data demonstrated comparable performance between the two modeling architectures, strongly suggesting that, at least for the current data and data volumes, a simple network architecture, like the one we have proposed here, is sufficient.

A critical issue for the development of machine learning models is the availability of accurate negative data. Often, not simply more but rather more accurate data are needed. Earlier works have proposed to resolve this issue by either mispairing the positive data, or by including data from healthy controls as negatives[14,15]. Both approaches share potential pitfalls in that the proposed negatives either share a compositional bias (imposed by the fact that they are positive to one or more of the other peptides in the data) or that the TCRs are falsely labeled as negative (imposed by the fact that TCRs in healthy controls are likely positive to the dominant peptides in the positive data set). Here, we have therefore taken a different approach, benefitting from the study published by 10X Genomics, and complemented the mis-paired artificial negative data with TCRs explicitly found not to be positive to any of the peptides in the training data. While this proved a highly useful approach, the 10X Genomics MHC-feature barcode platform is still in development, and the negative data defined here are hence likely not fully accurate. Given this, we suggest that substantial further work is needed to assess how to best define a proper TCR-negative data set.

The high performance of the developed NetTCR-2.0 model was validated on an in-house data set of paired TCR data with qualitative-interaction measurements to a set of 3 HLA-A*02:01 peptides. Here, a predictive positive value of ~75% was observed, greatly surpassing the performance of both the baseline and ERGO models. This result confirmed that the development of accurate prediction models for TCR specificity is contingent on the availability of paired (and accurate) α- and β-sequence data and suggests that a predictive power can be achieved to a degree where the tool can have actual biomedical applications.

Finally, in this work, we have used a rather simple definition of TCR similarity based on the relative Levenshtein distance when defining data redundancy. This distance has obvious short-comings when comparing the similarity between pairs of TCR of very different lengths—i.e., a similarity score of 0.9 corresponds to both one mutation/edit when comparing two TCRs of length 10 and to 4 mutations/edits if the TCRs are of length 36. Given the relatively limited length variation of the CDR3 sequences included in the current work (90% of the paired CDR3α and CDR3β sequences from the paired data set have a length in the range of 9–13 amino acids), this shortcoming does not have large impacts for the current work. However, it will be essential to consider alternative and less length-biased approaches, such as, for instance, the kernel similarity method underlying TCRmatch[22], if the work is extended to cover full-length TCRs and/or include the complete set of CDR sequences.

In conclusion, we have successfully trained a model to predict interactions between TCRs and their cognate, HLA-A*02:01-restricted peptide target. Our results indicate that accurate prediction is feasible only by training on data of paired TCRα- and β-chains. Due to the small number of training peptides, the model can at present only be applied to the limited set of peptides included in the training data. However, as more data become available, we expect the predictive power of the model to increase and allow for accurate predictions also for uncharacterized peptides, as has been observed earlier for the pan-specific prediction models of peptide-MHC interactions[30]. Finally, the presented model framework is highly flexible and allows for the straight-forward integration of the MHC molecule or TCRα chain in the future when data become available, to train a truly global prediction method.

## Materials and methods
### Training data
*CDRβ data*. The initial set of CDR3β sequences binding to epitopes presented by HLA-A*02:01 with corresponding epitopes was collected from the Immune Epitope Database (IEDB) on January 29th, 2020. The original IEDB data set consisted of 25,300 data points with 21,855 unique CDR3β sequences and 675 unique peptides, covering both class-I and -II binders. Cross-reactive TCRs were excluded. Quality assessment and uniform CDR3β-sequence frame were ensured by applying a k-mer-based scoring method using a profile hidden Markov model (pHMM) to the data (see Supplementary Note 1 details). Following quality assurance, the IEDB data set specific for HLA-A*02:01 and peptides of length 9 consisted of 10,987 unique CDR3β sequences and 168 peptides.

Nonbinding peptide-CDR3β pairs were derived from 10X Genomics Chromium Single Cell Immune Profiling of four donors. All T cells in this assay had been exposed to all tested pMHC multimers[31]. Each entry of the data set includes a unique molecular identifier (UMI) and counts of a given TCR to all peptides in the assay. From this data set, an initial negative data set was constructed from the HLA-A*02:01-restricted peptides filtered to only include TCR-peptide pairs with UMI counts <= 10. This data set comprised 1,325,949 distinct peptide-CDR3β pairs with 69,847 unique CDR3β sequences and 19 different peptides of which seven were shared with the IEDB peptides.

Positive and negative training data points were reduced to peptide-TCR pairs with CDR3β lengths within the range of 8–18 amino acids, and peptides of length equal to nine amino acids shared between the two data sets (7 peptides). The final data set representing seven epitopes characterized with both positive and negative TCR data consists of a positive set of 9204 unique CDR3β-peptide pairs and a negative data pool of 387,598 data points.

*Paired CDR data.* Positive data points were taken from IEDB and VDJdb. The databases were downloaded on August 26th, 2020 and August 5th, 2020, respectively. Restricting to data with both CDR3α and CDR3β chains available, a length range of 8–18 and reported to bind peptides of length 9, 3859 unique binding pairs were identified from IEDB and 2843 from VDJdb. These provided 4598 unique CDR3α-/β-peptide interactions with 276 different peptides specific to allele HLA-A*02:01.

Negatives were derived from 10X. Using the same restrictions as for the positives (CDR3 length between 8 and 18 AAs, peptide length 9, and peptides specific for HLA-A*02:01), 627,323 unique data points with 0 UMI counts to all the tested peptides were identified. These contained 33,017 unique TCRs tested against a set of 19 different peptides. In total, 17 of these overlapped with the peptides in the positive data set.

### External evaluation data

*MIRA.* Positive data points for external evaluation were derived from the MIRA set[32]. It entailed 376 CDR3β-peptide pairs associated with HLA-A*02:01. Negative samples were taken from an excluded subset of the 10X negative set (see above).

*Validation data.* Healthy donor material was collected under approval by the local Scientific Ethics Committee and written informed consent was obtained according to the Declaration of Helsinki. Peripheral blood mononuclear cells (PBMCs) from healthy donors were isolated from whole blood by density centrifugation on Lymphoprep (Axis-Shield PoC) and cryopreserved at −150 °C in FCS (FCS; Gibco) +10% DMSO.

The three peptides, GILGFVFTL, NLVPMVATV, and GLCTLVAML, were purchased from Pepscan (Pepscan Presto) and dissolved to 10 mM in DMSO. UV-sensitive ligands were synthesized as previously described[33]. In brief, recombinant HLA-A*02:01 heavy chains and human $\beta_2$ microglobulin light chain were produced in Escherichia coli. HLA heavy and light chains were refolded with UV-sensitive ligands. Specific peptide-MHC complexes were generated by UV-mediated peptide exchange[33] and MHC tetramers were assembled on PE-conjugated streptavidin (BioLegend, Nordic Biosite, Denmark) as previously described[34].

Cryopreserved PBMCs from four HLA-A*02:01-positive donors were thawed and washed in RPMI + 10% FCS. The presence of T cells binding to GILGFVFTL, NLVPMVATV, and GLCTLVAML was preestablished using DNA barcode-labeled MHC multimers as described in Bentzen et al.[9]. In total, $3 \times 10^6 - 6 \times 10^6$ cells from each donor were washed in cytometry buffer (PBS + 2% FCS) and incubated, 15 min, 37 °C, with a pool containing all three MHC multimers in a total volume of 80 μL (final concentration of each distinct pMHC, 23 nM). Next, a 5x antibody mix composed of CD8-BV480 (clone RPA-T8, BD 566121) (final dilution 1/50), dump-channel antibodies: CD4-FITC (BD 345768) (final dilution 1/80), CD14-FITC (BD 345784) (final dilution 1/32), CD19-FITC (BD 345776) (final dilution 1/16), CD40-FITC (Serotech MCA1590F) (final dilution 1/40), CD16-FITC (BD 335035) (final dilution 1/64), and a dead-cell marker (LIVE/DEAD Fixable Near-IR; Invitrogen L10119) (final dilution 1/1000) was added and incubated for 30 min at 4 °C. Cells were washed twice in cytometry buffer before proceeding directly to sorting.

Cells were sorted on a FACSMelody Cell Sorter (Becton Dickinson) into tubes containing 150 μl of PBS + 0.5% BSA (tubes were presaturated with PBS + 2% BSA). Using BD FACSChorus Software, we gated on single, live CD8-positive and "dump" (CD4, 14, 16, 19, and 40) negative lymphocytes. Within this population, we sorted all multimer-(PE) positive cells from all donors into one tube and a proportion of multimer negative/CD8 positive from all donors into another tube. The sorted cells were centrifuged for 10 min at 390 g and the buffer was removed. An overview of samples and gating strategy is included in Supplementary Table 1 and Supplementary Fig. 8.

VDJ sequences from the CD8 T cells were obtained through the 10x Genomics pipeline using Chromium Next GEM Single Cell 5′ Reagent Kits v2 (Dual Index) according to the manufacturer's instructions (10x Genomics, USA). Up to 17,000 cells of the multimer-positive or the multimer-negative CD8 T cells were loaded onto each of their separate lane, to yield a maximum of 10,000 cells with an intermediate/high doublet rate. TCRs were sequenced on a MiSeq as recommended by Illumina.

The single-cell data were processed via the 10x Genomics software Cell Ranger v5.0.1, using cellranger mkfastq and cellranger vdj, to extract V(D)J gene annotations and CDR3 sequences for each T cell. The GRCh38/Ensembl reference genome v4.0.0 for mapping V(D)J genes was downloaded from 10x Genomics. The pool of all multimer-positive cells and the pool of multimer-negative cells yielded 1091 and 12,801 mapped and annotated T cells, respectively. Of these sets, 520 and 3074 cells, respectively, met the criteria of having both an α- and β-chain with unambiguous annotations, meaning that each T cell should only have one α-chain and one β-chain annotation. Reducing the sets to contain only unique pairs of CDR3 α/β and removing the TCRs already present in the training set, resulted in 89 multimer-positive pairs and 1694 multimer-negative pairs.

*Data preparation.* Figure 9 gives a schematic overview of how the data-redundancy and data partitioning procedure was implemented in the current work. The sections below describe the details of each of the outlined steps.

*Similarity scoring.* A critical component of data redundancy is related to the metric chosen to define the similarity between two points. Here, the Levenshtein similarity was used as a measure of the similarity between CDR3 sequences. The Levenshtein similarity is based on the Levenshtein distance. The Levenshtein distance is a similarity measure between words. Given two strings, the distance describes the number of modifications needed to transform one word into another. The possible changes are insertion, deletion, and replacement. Each of these three operations adds one to the distance. The Levenshtein similarity score is given by the relation

$$Sim_{Lev} = \frac{max(|u|, |v|) - Distance_{Lev}(u, v)}{max(|u|, |v|)}, \quad (1)$$

where $u$ and $v$ represent two CDR3 sequences, and $|\cdot|$ defines their length.

*Redundancy reduction.* Peptide-specific redundancies regarding CDR3 sequences were removed using the Hobohm 1 algorithm[35]. The positive and negative data specific for each peptide were each first sorted by CDR3 length in descending order. Next, the sorted negative data were appended to the sorted positive data. Sequences were then iteratively sorted into non-redundant and redundant stacks based on a given similarity threshold, hereafter referred to as redundancy threshold. The algorithm starts by assigning the first sequence to the nonredundant list. It then iterates through the peptide-specific CDR3 sequences and assesses whether a sequence's similarity to the list of nonredundant sequences is above the redundancy threshold or not. Similarities above the threshold lead to the examined sequence being assigned to the redundant list.

*Data partitioning.* Partitioning was performed using single-linkage clustering of the redundancy-reduced positive training data. First, the Levenshtein similarity scores between all CDR3 sequences are binarized based on a given threshold, referred to as the data-partitioning threshold. In the case of paired-chain data, TCR similarity is defined as the average α and β Levenshtein similarity. Next, single-linkage clustering was performed on this binary matrix, and the connected components of this graph were sorted by size into a list and iteratively assigned partitions 1–5. The selected similarity threshold thus presents an upper limit of similarities between different partitions.

Next, negative CDR3 data were added to each partition. For each peptide in each partition, 5 times the number of positive CDR3 were added from the negative data. Negatives were gradually added under the condition that their similarity to all TCRs in the other partitions was lower than the given partitioning threshold. In addition, negative examples were generated by mismatching the positive data, i.e., combining a TCR sequence with a peptide different from its cognate target. Each positive TCR was paired with 5 peptides, randomly sampled from the list of unique peptides in the dataset. These added negatives were used during the training but were not included when evaluating the model performance.

*Separating external Evaluation Data from the Training Data.* The evaluation data sets for the CDR3β model were separated from the training data by a given Levenshtein similarity threshold, meaning that the data points with similarities to the training data above this threshold were removed. Negatives reserved for external evaluation were reduced to CDR3β sequences with similarities below the given threshold to the training data. Subsequently, five times the number of positives per peptide were randomly selected from the remaining negatives.

*Paired chain data preparation pipeline.* Positive and negative data from IEDB, VDJdb, and 10X were prepared and cleaned as described in the training data section. Positives and negatives were then reduced to data points containing their shared set of peptides. This is represented by 18 different peptides and resulted in 2886 unique positive interactions and 594,306 unique negative data points. Positive data were subsequently partitioned into 5 partitions with a similarity threshold based on their average chain similarities. Negatives were then added to the partitioned positives as 5 times the number of positives per peptide and partition, upholding the similarity restraint of the partitioning. Further were mismatched negatives added as described above.

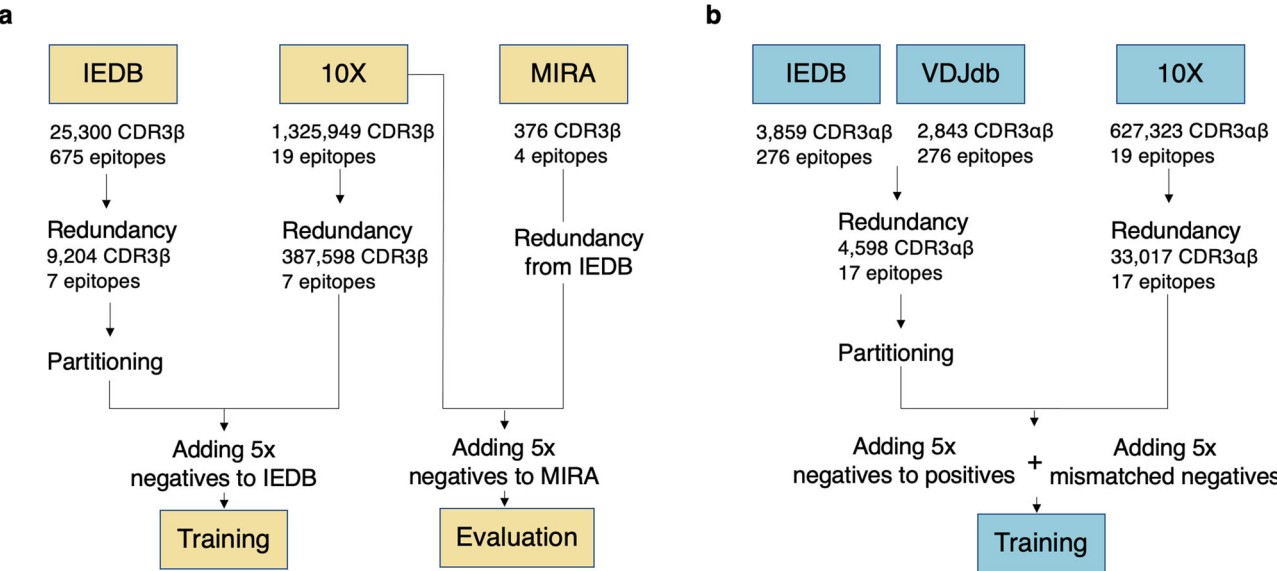

**Fig. 9 Data-partitioning pipeline schematics. a** Data-preparation pipeline for the β-chain data; **b** pipeline for the paired-chain data. The positive and negative data sets were each redundancy-reduced with the Hobohm 1 algorithm, according to a Levenshtein similarity threshold. The redundancy-reduced set of positives was partitioned into five groups using a single-linkage clustering algorithm. Negative data were subsequently added to each partition: for each peptide, 5 times the number of positives was randomly selected from the pool of nonredundant negative data. In **a**, to ensure that the MIRA external evaluation data did not share similarity with the training set, positive points from the MIRA set with a Levenshtein similarity above a certain threshold were removed. Each step of the pipeline is described in detail in the text.

*Baseline model.* A baseline model was designed to establish the predictive power of simple similarity-based methods. The similarity-scoring approach used in the baseline model was the kernel-scoring method introduced by Shen et al.[36] with default parameters, as described earlier in the MAIT Match[19] and TCRMatch[22] methods. In the model, the prediction score for a given TCR is calculated as the highest score obtained when scoring the CDR3β against a database of positive CDR3βs. In 5-fold cross-validation, each of the 5 partitions, in turn, represents a test set, and the positive elements in the remaining 4 partitions define the database. For external evaluation, all positive elements in the training data set define the database. For analysis of paired α and β TCR sequences, the similarity score was calculated as the highest average of the individual α and β CDR3-sequence scores for each TCR.

*TCRdist model.* The TCRdist model was implemented identically to the baseline model only using the distance metric proposed in the TCRdist publication[7]. That is, the prediction score for a given TCR is calculated as 1—the closest distance obtained when scoring the TCR against a database of positive TCRs for the given peptide (defined in a cross-validated manner).

**Neural networks**

*The NetTCR model.* A 1-dimensional CNN model, similar to the one proposed by Jurtz et al.[15], was implemented to predict whether or not a given TCR can bind to a specific peptide. The neural network takes the peptide, the CDR3α, and/or CDR3β regions of the TCR amino acid sequences as inputs. The CDR sequences were zero-padded to a maximum length of 30. The amino acids were encoded using the BLOSUM50 matrix[37]. That is, each amino acid is represented as the score for substituting the amino acid with all the 20 amino acids. Hence, the BLOSUM encoding scheme maps a sequence of length $l$ into an array of dimension $l$ x 20. The peptide and the CDR3 sequences are processed separately by a 1D convolutional layer with channels corresponding to the given sequence encoding. On each sequence (peptide, CDR3(s)), 16 convolutional filters with kernel size {1, 3, 5, 7, 9} process the input (80 filters per sequence). The kernel weights were initialized with the Glorot normal initializer[38]. For each kernel size, the convolutional output was max-pooled and the resulting feature vectors concatenated in a single vector with 240 entries (80 for each input sequence) representing the convoluted peptide and CDR3 sequences. This vector was then fed into a dense layer of 32 hidden neurons; the output consists of one single neuron, giving the probability of a peptide-TCR pair to bind. The activation function used through the network was the sigmoid function. A schematic representation of the CNN model is given in Fig. 10.

*Model training.* Models were trained using nested 5-fold cross-validation (CV) for 300 epochs with early stopping and patience of 50 epochs. The weights were updated using the Adam optimizer with a learning rate of 0.001. The batch size was 128 and the loss function was binary cross-entropy.

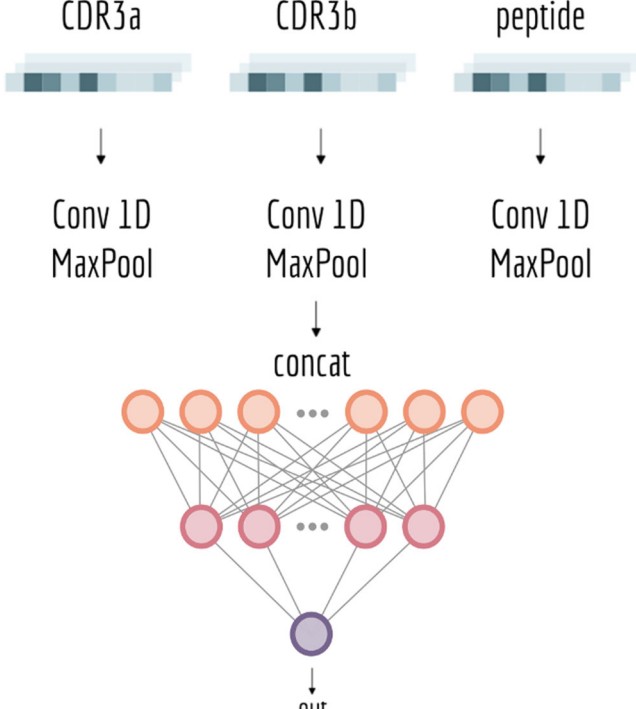

**Fig. 10 Setup of NetTCR model.** The CDR3 and peptide sequences are encoded using the BLOSUM50 matrix. The encoded sequences are passed independently through a 1D convolutional layer and a max-pooling layer. The convolutional filter size is set to {1, 3, 5, 7, 9}, and for each filter size, 16 filters are used. The extracted features are then concatenated and fed into a dense layer with 32 hidden units. The output of the network consists of a single neuron, giving the binding probability.

*Performance evaluation.* In cross-validation, the performance was evaluated from the concatenated test sets either globally over the entire data set, or in a per-peptide manner. Likewise was the performance on the independent evaluation reported either globally over the entire data set, or in a per-peptide manner. To normalize the prediction scores across peptides, the raw prediction values were transformed into the percentile rank values. Percentile rank scores were estimated from a set of 10,000 natural TCRs, extracted from the 10X data set with no overlap with the training set. The percentile rank score of a given peptide-TCR pair was then calculated by comparing the prediction score with the distribution of prediction scores for the particular peptide.

**Reporting summary**. Further information on research design is available in the Nature Research Reporting Summary linked to this article.

## Data availability
All data and data partitions used for NetTCR-2.0 training and evaluation are available at https://github.com/mnielLab/NetTCR-2.0.

## Code availability
The NetTCR-2.0 code is available at https://github.com/mnielLab/NetTCR-2.0. The NetTCR-2.0 prediction model is available as a web-server tool at https://services.healthtech.dtu.dk/service.php?NetTCR-2.0.

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

## Acknowledgements
We would like to thank DTU Multi-Assay Core (DMAC) for sequencing the set of novel paired-chain TCRs. This research was funded in part through the Independent research fund Denmark (DFF-7014-00055 to M.N), the Federal funds from the National Institute of Allergy and Infectious Diseases, National Institutes of Health, Department of Health and Human Services (under Contract No. HHSN272201200010CERC to M.N), StG NextDART (677268 to S.R.H.), and the Lundbeck Foundation Experiment (R324-2019-1671 to A.K.B.).

## Author contributions
M.N., L.E.J., A.M., V.S. and V.J. designed the study. A.M. and V.S. conducted the majority of the experiments. V.S. and L.E.J. contributed to the data cleaning and partitioning pipeline. A.K.B., H.R.P. and S.R.H. performed the T.C.R. sequencing and analysis. W.D.C., A.C. and B.J. contributed to the data collection and method-performance comparisons. The paper was written by A.M., V.S., L.E.J. and M.N. with contributions from all authors. All authors have read and approved the final version of the paper.

## Competing interests
The authors declare no competing interests.
