## [Peer Review File · Communications Biology]

NetTCR-2.0 enables accurate prediction of TCR-peptide binding by using paired TCR α and β sequence dataReviewers' comments:

Reviewer #1 (Remarks to the Author):

In this manuscript by Montemurro and colleagues, the authors have addressed the challenging issue of TCR-pMHC prediction. The authors have evaluated existing predictors of TCR-pMHC interactions, a field facing three challenges as discussed by the authors: data accuracy, data scarceness and problem complexity. The authors show that low quality of current publicly available data limits the development of accurate predictors. The authors have investigated several model architectures and concluded that a simple 1D convolutional neural network-based model was the overall best performing model, outperforming sequence-based models.

The main conclusion of the study is that interactions between TCR and cognate pMHC can be predicted pending a large number of paired data are available (evaluated as > 150). This means that current tools can only predict interaction with pMHC used in the "training" dataset. The authors acknowledge that this is a limitation in particular for the emerging field of personalized cancer immunotherapy.

The internal validation was performed using a dataset of 3 virus-specific TCR obtained from 3 healthy donors. Given the limited number of distinct pMHC, it would be interesting to understand whether TCR clustering algorithms such as Ostmeier Cancer research 79, 1671-1680 (2019), Glanville Nature 547, 94-98 (2017) or Dash Nature 547, 89-93 (2017) would contribute to increase overall performance.

Reviewer #3 (Remarks to the Author):

The manuscript "NetTCR-2.0: Accurate Prediction of TCR-Peptide Binding is Contingent on Paired TCR α and β Sequence Data" by Montemurro et al. extends on their previously reported machine learning approach for predicting TCR specificity for a given peptide-MHC combination by evaluating the effect on performance when paired data was considered. Here, they show training the NetTCR model with paired TCR information improved the sensitivity of the predictions compared to one trained on TCR beta chain information alone. Further benchmarking their model, they show it performed better than their similarity-based baseline model and comparable to a more complex one, showing that a relatively simple sequence-based model is capable of accurately predicting whether a TCR will bind a given epitope. While the conclusion that training with paired TCR data is better than single chain data at predicting specificity is not too surprising, this point often is ignored in the field and more cases like this help bring attention to the need for higher quality datasets for this purpose. An important topic discussed by the authors is the lack of "negative" TCR data available for accurately training predictive models generally in the field and they take some important steps to address this limitation. My major concern is the model was trained only a small number of uniform peptides in the context of one MHC molecule, and with the repertoires for their three main peptides of have been relatively deeply characterized it is hard to determine how well it extends across varying MHCs and peptide lengths.

Major comments

Figure 1 is helpful for describing how the datasets were cleaned but listing the information about the data points (ie. Dataset, epitope, number data points) used for training and evaluation in table would be useful.

Only TCR pairs binding to epitopes that are 9 amino acids in length were chosen for the training data I assume because it is simpler if they are uniform in length but the reason for that decision is not explained well despite being a potential limitation to the model.

Why was only HLA-A*02:01 chosen? A number of the pMHCs in the 10x datasets clearly show non-specific binding and should obviously not be included, but there are many others in the dataset (e.g. the HLA-B epitopes) that are more specific. Is it due to the peptide length restriction?

Given the limited number of peptides examined, how much does the peptide sequence CNN layer really contribute to the model's performance?

Negative CDR3 data is sorted into the peptide-specific CDR3 partitions in the training set based on similarity. TCR gene usage is important in shaping the convergent TCRs recognizing the same pMHC. How well-matched is the VDJ usage between the binders and the non-binders?

Line 376 suggests the quality of the IEDB data compared to the MIRA data accounts for the difference in the models' performance between the CV and MIRA datasets. Is there a reason for excluding VDJdb from the single-chain training dataset? Figure 5 shows that training ERGO with datasets from different repositories highlights how the data quality/heterogeneity/depth affects performance, and this should be similarly explored for NetTCR.

Minor Comments

Line 79. "They present a wide range of data used and applied modeling techniques " confusing wording.

Line 88. NLP not defined

Line 117. Spelling "assay"

Title in figure 6c should be 'partitioning', I believe.

Unclear which figure lines 474-490 is referring to.

The legend for Figure 8 is overly complicated.

This statement on line 493 is unclear, "Here, binding to the three peptides GIL, NLV, and GLC was performed (using the-validated predictions) for each TCR positive to any of these peptides."

Line 575. What exactly is AUC0.1, the threshold? PPV is not defined.

Reviewer #4 (Remarks to the Author):

The manuscript describes a new ML-based tool that predicts peptide binding to T-cell receptors, highlighting the need for using both alpha and beta chains of TCRs. The research is highly relevant and novel, and the resulting model is made available in a web server for community use. Interestingly, the authors show that a (relatively) simple model, a 1D convolutional NN, can outperform complex models when using richer data sets composed of paired alpha- and beta-chains.

This is a continuation of previous research which published the first NetTCR model. Authors do a laudable effort to create a easy user interface, with detailed instructions, which facilitates the use of the web interface and could lead to broad use within this research community. Similarly, authors released all training and testing data in a version controlled repository.

This is a timely manuscript that expands the available tools in TCR-pMHC context. Moreover, it exemplifies the importance of data gathering and processing when training ML models, and proves that a simpler model may outperform a more computationally expensive one when the data is available to do so. It also makes exemplary use of statistical analysis, bootstrapping, and experimental controls to test, compare and quantify model performance.

Major issues:

The only major issue in this manuscript is the lack of methodological detail that prevents an exact reproduction of the research. Despite the laudable effort to publish training and testing data, and carefully explain the data gathering, selection, and splitting methods, many details are missing, and no code is provided to reproduce the NN training process.

For example,

- what is the actual Levenshtein similarity threshold used to partition the data (line 237) in the server?

- how is BLOSUM, a similarity matrix, used to encode amino acids, as indicated in line 294?
- how were NN weights initialized (not mentioned in section Model Training, line 313)?

There are many details to creating and training a NN that can drastically impact its performance, so an actual python script that shows how the Keras API was used is essential to ensure reproducibility. Similarly to scikit models. (assuming these were the only ones, as indicated in the reporting summary).

Minor issues:

The application presented in Figure 7, to rank peptides that could bind a given TCR a/b sequence pair, does not seem to be available in the web server, or in stand-alone code. This could be a highly generalizable approach to design epitopes, and of great interest to the community. Why is it not available for use to the readers?

On Figure 2, given the simplicity of the model, it would be easier to understand the model if the network depicted there had the actual architecture. As it stands, the figure has NN with two hidden layers of with 7 and 4 neurons each. Could the figure have a single 32-neuron layer that matches the network described in the text?

Since the title of the paper indicates that a NetTRC version 1 existed, the reader could better understand the novelty presented in this work if the authors more explicitly differentiated version 1 from the current version 2.0, and highlighted the improvements made to the method.

Line 371: The language is confusing when describing Figure 4B. It is claimed that the model was trained on a data set partitioned using a 94% similarity threshold, but the figure has an X-axis with MIRA thresholds. Please clarify what the X axis refers to and what is the difference between the threshold being mentioned.

Line 374: Clarify what is the performance metric presented in "...performance value of up to 0.79."

Line 412: Which two models?

Line 79: The sentence "They present a wide range of data used and applied modelling techniques" seems to be missing a verb.

Line 90: Should read "limits".

Not all SI figures have titles. This should be standardized.

We thank the reviewers for their valuable time and comments. In the following pages, we answer (highlighted in blue) point-by-point to the reviewers' comments and concerns, and describe the changes to the manuscript (highlighted also in blue the revised manuscript).

Reviewers' comments:

Reviewer #1 (Remarks to the Author):

In this manuscript by Montemurro and colleagues, the authors have addressed the challenging issue of TCR-pMHC prediction. The authors have evaluated existing predictors of TCR-pMHC interactions, a field facing three challenges as discussed by the authors: data accuracy, data scarceness and problem complexity. The authors show that low quality of current publicly available data limits the development of accurate predictors. The authors have investigated several model architectures and concluded that a simple 1D convolutional neural network-based model was the overall best performing model, outperforming sequence-based models.

The main conclusion of the study is that interactions between TCR and cognate pMHC can be predicted pending a large number of paired data are available (evaluated as > 150). This means that current tools can only predict interaction with pMHC used in the "training" dataset. The authors acknowledge that this is a limitation in particular for the emerging field of personalized cancer immunotherapy.

The internal validation was performed using a dataset of 3 virus-specific TCR obtained from 3 healthy donors. Given the limited number of distinct pMHC, it would be interesting to understand whether TCR clustering algorithms such as Ostmeyer Cancer research 79, 1671-1680 (2019), Glanville Nature 547, 94-98 (2017) or Dash Nature 547, 89-93 (2017) would contribute to increase overall performance.

We believe that the approaches suggested by Ostmeyer and Glanville are different from what we are suggesting with NetTCR, and these methods cannot directly be applied to predict TCR specificity. However, we agree that the TCRdist method published by Dash can be used to predict TCR-peptide binding, and we have included a comparison between NetTCR-2.0 and TCRdist (Dash et al, 2017) in the revised manuscript.

Reviewer #3 (Remarks to the Author):

The manuscript “NetTCR-2.0: Accurate Prediction of TCR-Peptide Binding is Contingent on Paired TCR α and β Sequence Data” by Montemurro et al. extends on their previously reported machine learning approach for predicting TCR specificity for a given peptide-MHC combination by evaluating the effect on performance when paired data was considered. Here, they show training the NetTCR model with paired TCR information improved the sensitivity of the predictions compared to one trained on TCR beta chain information alone. Further benchmarking their model, they show it performed better than their similarity-based baseline model and comparable to a more complex one, showing that a relatively simple sequence-based model is capable of accurately predicting whether a TCR will bind a given epitope. While the conclusion that training with paired TCR data is better than single chain data at predicting specificity is not too surprising, this point often is ignored in the field and more cases like this help bring attention to the need for higher quality datasets for this purpose. An important topic discussed by the authors is the lack of “negative” TCR data available for accurately training predictive models generally in the field and they take some important steps to address this limitation. My major concern is the model was trained only a small number of uniform peptides in the context of one MHC molecule, and with the repertoires for their three main peptides of have been relatively deeply characterized it is hard to determine how well it extends across varying MHCs and peptide lengths.

Major comments

Figure 1 is helpful for describing how the datasets were cleaned but listing the information about the data points (ie. Dataset, epitope, number data points) used for training and evaluation in table would be useful.

We agree and have updated Figure 1 to include the numbers of TCR sequences and epitopes at each step of the data preparation pipeline.

Only TCR pairs binding to epitopes that are 9 amino acids in length were chosen for the training data I assume because it is simpler if they are uniform in length but the reason for that decision is not explained well despite being a potential limitation to the model.

Why was only HLA-A*02:01 chosen? A number of the pMHCs in the 10x datasets clearly show non-specific binding and should obviously not be included, but there are many others in the dataset (e.g. the HLA-B epitopes) that are more specific. Is it due to the peptide length restriction?

In this study, we have used the 10X study only to define negative data. As a resource for positive data, we used IEDB (and VDJdb), and the vast majority of these data are specific for HLA-A02:01. Further was the purpose of this study to investigate the impact of data limitations/quality and integration of paired TCR sequence information on the model performance. Given this, we have sought to keep the modeling complexity of the problem to a minimum. The same argument applies to why we have limited the model to only include 9mer peptides (defining more than 98% of the positive IEDB data set). We are aware that this is a limitation of the current study, and mention this in the discussion of the manuscript.

Given the limited number of peptides examined, how much does the peptide sequence CNN layer really contribute to the model's performance?

This is an important question, and we believe to have addressed this in figure 8 (the original figure 7), where we demonstrate how the model can accurately predict the correct peptide target for a given positive TCR. Such performance values could only be obtained from a peptide specific model. However to further quantify to what extent the peptide sequence contributes to the overall performance, we trained a model where the TCRab sequences were paired with a wrong peptide. Next, we ran the peptide ranking analysis with these models. The figure below shows the proportion of TCRs for which the predicted lowest-ranking peptide matched with the "true" target peptide (equivalent to Figure 8 in the manuscript). The figure shows that, for instance, NetTCR_ab will always assign the lowest rank score to the GIL peptide, hence only that peptide will be selected. This shows that the models (trained with the wrong peptides) have not learnt to pair the TCRs with the correct peptide. We have updated the manuscript to include a brief

description of these results.

Negative CDR3 data is sorted into the peptide-specific CDR3 partitions in the training set based on similarity. TCR gene usage is important in shaping the convergent TCRs recognizing the same pMHC. How well-matched is the VDJ usage between the binders and the non-binders?

We agree that biases in VDJ usage between the positive and negative data could influence the model performance, and also that we in the manuscript have not explicitly investigated this. One key reason for this is that VDJ gene annotation is only available for a limited set of our training data (IEDB does only to a very limited degree include this information). We expect this will change with future updates to the IEDB, and clearly intend to investigate this topic more and include VDJ /and/or CDR1 and 2) information in future updates to the NetTCR modeling framework. We have mentioned this in the discussion, and stated that “future work will tell if integrating this information can lead to an improved predictive power of the model proposed here”.

Line 376 suggests the quality of the IEDB data compared to the MIRA data accounts for the difference in the models’ performance between the CV and MIRA datasets. Is there a reason for excluding VDJdb from the single-chain training dataset? Figure 5 shows that training ERGO with datasets from different repositories highlights how the data quality/heterogeneity/depth affects performance, and this should be similarly explored for NetTCR.

In this study, we included only positive CDR3b from the IEDB. We agree that including VDJdb data could potentially have expanded the data volume. However this expansion would have been minor due to the very large overlap between the two databases. At the time where the data for the study was extracted, the overlap between the two databases was > 90% (when considering HLA-A*02:01 9mer peptide specific TCR). Further, the main purpose of working with the CRD3b data was to assess to what degree such data had the quality and quantity to develop accurate prediction models. Given the clear results on this from the analysis on the IEDB data, including a minority data extension from VDJdb is not expected to alter this. Finally, and again due to the high data set overlap, and due to the results shown in figure 5 where the NetTCR and ERGO modeling frameworks are demonstrated to have comparable predictive power (Figure 5A), and the NetTCR model trained on IEDB data, and the ERGO data trained on VDJdb data show a comparable performance on the independent MIRA data (Figure 5B), we believe that comparing training of NetTCR with data from IEDB and VDJdb would add little if any novelty to the study. Further, we believe that the performance difference observed between ERGO model trained on McPAS and VDJdb is due to data quantity rather than data quality, since the VDJdb training data is almost 4 times larger than the McPAS data.

Minor Comments

Line 79. “They present a wide range of data used and applied modeling techniques “ confusing wording.

This sentence has been rephrased.

Line 88. NLP not defined

We clarified that.

Line 117. Spelling “assay”

We corrected that.

Title in figure 6c should be ‘partitioning’, I believe.

The label 'partitioning' is not the title of Figure 6C but the x-axis label of figure 6B. The labels have been removed.

Unclear which figure lines 474-490 is referring to.

There is no figure for these results. In the text we only report the AUC values obtained from the experiment.

The legend for Figure 8 is overly complicated.

We have changed the legend for Figure 8, and removed the colour gradient for the circles (representing the prediction scores).

This statement on line 493 is unclear, “Here, binding to the three peptides GIL, NLV, and GLC was performed (using the-validated predictions) for each TCR positive to any of these peptides.”

This has been corrected

Line 575. What exactly is AUC0.1, the threshold? PPV is not defined.

This part has been rewritten, defining all the performance measures used.

Reviewer #4 (Remarks to the Author):

The manuscript describes a new ML-based tool that predicts peptide binding to T-cell receptors, highlighting the need for using both alpha and beta chains of TCRs. The research is highly relevant and novel, and the resulting model is made available in a web server for community use.

Interestingly, the authors show that a (relatively) simple model, a 1D convolutional NN, can outperform complex models when using richer data sets composed of paired alpha- and beta-chains.

This is a continuation of previous research which published the first NetTCR model. Authors do a laudable effort to create a easy user interface, with detailed instructions, which facilitates the use of the web interface and could lead to broad use within this research community. Similarly, authors released all training and testing data in a version controlled repository.

This is a timely manuscript that expands the available tools in TCR-pMHC context. Moreover, it exemplifies the importance of data gathering and processing when training ML models, and proves that a simpler model may

outperform a more computationally expensive one when the data is available to do so. It also makes exemplary use of statistical analysis, bootstrapping, and experimental controls to test, compare and quantify model performance.

Major issues:

The only major issue in this manuscript is the lack of methodological detail that prevents an exact reproduction of the research. Despite the laudable effort to publish training and testing data, and carefully explain the data gathering, selection, and splitting methods, many details are missing, and no code is provided to reproduce the NN training process.

For example,

- what is the actual Levenshtein similarity threshold used to partition the data (line 237) in the server?

We have now specified the models we use in the server.

- how is BLOSUM, a similarity matrix, used to encode amino acids, as indicated in line 294?

We have clarified how the BLOSUM matrix is used for encoding the sequences.

- how were NN weights initialized (not mentioned in section Model Training, line 313)?

We have included some more details about the network.

There are many details to creating and training a NN that can drastically impact its performance, so an actual python script that shows how the Keras API was used is essential to ensure reproducibility. Similarly to scikit models. (assuming these were the only ones, as indicated in the reporting summary).

The GitHub repo has been updated (<https://github.com/mnielLab/NetTCR-2.0>). It now contains a python script with the core code for NetTCR method.

Minor issues:

The application presented in Figure 7, to rank peptides that could bind a given TCR a/b sequence pair, does not seem to be available in the web server, or in stand-alone code. This could be a highly generalizable approach to design epitopes, and of great interest to the community. Why is it not available for use to the readers?

This option is available to the user. The NetTCR server output has a column called "percentile_rank". This column has been used to make Figure 8 (old figure 7) and can be used to rank the peptides binding to TCRs. One should pick the peptide that has the lowest percentile_rank.

On Figure 2, given the simplicity of the model, it would be easier to understand the model if the network depicted there had the actual architecture. As it stands, the figure has NN with two hidden layers of with 7 and 4 neurons each. Could the figure have a single 32-neuron layer that matches the network described in the text?

We believe including a 32 hidden neuron layer would over-complicate the figure. Instead, we have edited Figure 2, so that the number of hidden neurons in the figure should not be ambiguous.

Since the title of the paper indicates that a NetTRC version 1 existed, the reader could better understand the novelty presented in this work if the authors more explicitly differentiated version 1 from the current version 2.0, and highlighted the improvements made to the method.

We have added a discussion about NetTCR-1.0, stating the differences with the current version and explaining what NetTCR-2.0 performs better.

Line 371: The language is confusing when describing Figure 4B. It is claimed that the model was trained on a data set partitioned using a 94% similarity threshold, but the figure has an X-axis with MIRA thresholds. Please clarify what the X axis refers to and what is the difference between the threshold being mentioned.

We have rephrased the caption in Figure 4B, clearly stating what are the thresholds used.

Line 374: Clarify what is the performance metric presented in "...performance value of up to 0.79."

Has been corrected.

Line 412: Which two models?

Has been clarified.

Line 79: The sentence “They present a wide range of data used and applied modelling techniques” seems to be missing a verb.

Has been corrected.

Line 90: Should read “limits”.

Has been corrected.

Not all SI figures have titles. This should be standardized

It is not clear to us what the reviewer is referring to here. None of the SI figures have titles.

REVIEWERS' COMMENTS:

Reviewer #3 (Remarks to the Author):

The number of non-redundant TCRs noted in figure 1B seems incorrect. It's greater than the total number it came from and it is identical to the number in figure 1A.

Otherwise, the author's revisions and rebuttals have addressed my original concerns and I have no further ones.

Reviewer #4 (Remarks to the Author):

The authors have responded all points sufficiently for me to support the publication of this manuscript.

I believe the extra material added to the text helps to explain the methodological details and will help the manuscript be reproducible.

My one comment is that authors indicated they had added a discussion to better differentiate versions 1 and 2 of NetTCR, however that was not highlighted in the discussion section and I could not identify where this extended discussion was inserted in the manuscript.

A clear distinction and explanation of what was added to 2.0 will benefit the reader in understanding the advances presented in this work.

REVIEWERS' COMMENTS:

Reviewer #3 (Remarks to the Author):

The number of non-redundant TCRs noted in figure 1B seems incorrect. It's greater than the total number it came from and it is identical to the number in figure 1A.

Otherwise, the author's revisions and rebuttals have addressed my original concerns and I have no further ones.

We thank the reviewer for the comment. The numbers in Figure 1B were indeed wrong and now they have been corrected.

Reviewer #4 (Remarks to the Author):

The authors have responded all points sufficiently for me to support the publication of this manuscript.

I believe the extra material added to the text helps to explain the methodological details and will help the manuscript be reproducible.

My one comment is that authors indicated they had added a discussion to better differentiate versions 1 and 2 of NetTCR, however that was not highlighted in the discussion section and I could not identify where this extended discussion was inserted in the manuscript.

A clear distinction and explanation of what was added to 2.0 will benefit the reader in understanding the advances presented in this work.

We are sorry this update was unintentionally left out from the revised manuscript. We have now rewritten part of the introduction to clarify how version 2.0 is related to the earlier version 1.0.